Climatic and soil characteristics account for the genetic structure of the invasive cactus moth Cactoblastis cactorum, in its native range in Argentina

Andraca-Gómez Guadalupe 1 2 guadalupeandraca@gmail.com
http://orcid.org/0000-0003-0962-973X Ordano Mariano 3
http://orcid.org/0000-0002-3219-0019 Lira-Noriega Andrés 4
http://orcid.org/0000-0003-0701-5398 Osorio-Olvera Luis 2
Domínguez César A. 2
Fornoni Juan 2 jfornoni@iecologia.unam.mx
1 Instituto de Biología, Universidad Nacional Autónoma de México , Ciudad Universitaria, Ciudad de México , México
2 Instituto de Ecología, Universidad Nacional Autónoma de México , Ciudad Universitaria, Ciudad de México , México
3 CONICET-UNT, Fundación Miguel Lillo-Instituto de Ecología Regional , San Miguel de Tucumán, Tucumán , Argentina
4 Instituto de Ecología, A.C., CONAHCYT Research Fellow , Xalapa, Veracrúz , México
Drobniak Szymon
Electronic publication date: 2024 Feb 12
Publication date: 2024
Volume: 12
Electronic Location ID: e16861
Received 2023 Oct 6; Accepted 2024 Jan 9
Copyright: © 2024 Andraca-Gómez et al.
Copyright year: 2024
Copyright holder: Andraca-Gómez et al.
License: This is an open access article distributed under the terms of the Creative Commons Attribution License, which permits unrestricted use, distribution, reproduction and adaptation in any medium and for any purpose provided that it is properly attributed. For attribution, the original author(s), title, publication source (PeerJ) and either DOI or URL of the article must be cited.
License URL: https://creativecommons.org/licenses/by/4.0/

Keywords: Biological invasions, Gene flow, Lepidoptera, Migration, Population genetics, Prickly pear cacti

Funding: CONABIO PAPIIT IN210922 Instituto de Biología (UNAM) from DGAPA-UNAM This work was financed by a CONABIO grant to Juan Fornoni, Karina Boege, and César A. Domínguez, and PAPIIT IN210922 to Juan Fornoni. Guadalupe Andraca-Gómez received a post-doctoral fellow at the Instituto de Biología (UNAM) from DGAPA-UNAM. The funders had no role in study design, data collection and analysis, decision to publish, or preparation of the manuscript.

==============================
Background

Knowledge of the physical and environmental conditions that may limit the migration of invasive species is crucial to assess the potential for expansion outside their native ranges. The cactus moth, Cactoblastis cactorum, is native to South America (Argentina, Paraguay, Uruguay and Brazil) and has been introduced and invaded the Caribbean and southern United States, among other regions. In North America there is an ongoing process of range expansion threatening cacti biodiversity of the genus Opuntia and the commercial profits of domesticated Opuntia ficus-indica.

Methods

To further understand what influences the distribution and genetic structure of this otherwise important threat to native and managed ecosystems, in the present study we combined ecological niche modeling and population genetic analyses to identify potential environmental barriers in the native region of Argentina. Samples were collected on the host with the wider distribution range, O. ficus-indica.

Results

Significant genetic structure was detected using 10 nuclear microsatellites and 24 sampling sites. At least six genetic groups delimited by mountain ranges, salt flats and wetlands were mainly located to the west of the Dry Chaco ecoregion. Niche modeling supports that this region has high environmental suitability where the upper soil temperature and humidity, soil carbon content and precipitation were the main environmental factors that explain the presence of the moth. Environmental filters such as the upper soil layer may be critical for pupal survival and consequently for the establishment of populations in new habitats, whereas the presence of available hosts is a necessary conditions for insect survival, upper soil and climatic characteristics will determine the opportunities for a successful establishment.

Introduction

Since Elton’s book on the Ecology of Invasions by Animals and Plants (Elton, 1958), the field of invasion biology has grown exponentially (Ricciardi & MacIsaac, 2008), but our ability to predict which physical and biotic factors will prevent the expansion of invasive species in their non-native range is still poorly developed (Richardson, 2011). So far, rates of invasion have increased during the last century despite control and management practices (Jaspers et al., 2021), suggesting that being able to predict the invasion dynamic will open new opportunities to cope this threat. A central element in predicting the potential migration of invasive species in foreign regions is the analysis of the natural barriers that define the spatial distribution in their native habitat (Sherpa et al., 2019). Thus, understanding native spatial patterns of dispersal of individuals and genes is a first line of evidence to identify potential environmental barriers as input for predictive models of invasion and population management.

The simplest hypothesis about gene flow establishes that this is mainly determined by the geographic distance that separates two or more populations (Isolation by Distance, IBD) (Wright, 1943). However, to find a pattern of IBD, it is necessary that the flow between populations is constant, that nothing interferes with the movement of genes in all directions (neither physical nor environmental barriers), and that other evolutionary processes like drift or selection are weaker than the intensity of gene flow (Bolnick & Nosil, 2007, Epperson, 2010). Also, the IBD analysis does not provide information on whether environmental factors are interacting with evolutionary processes (Manel et al., 2003). To identify how the environment can contribute to facilitate or reduce the rates of movement of genes between different populations, tools have been developed in recent years to analyze various gene flow hypotheses (Anderson et al., 2010). Circuit theory has been used to build testable hypotheses of gene flow based on the ecology of the species and the presence of potential environmental and physical barriers (e.g., McRae, Shah & Mohapatra, 2009; Andraca-Gómez et al., 2015; Dickson et al., 2019). This information is used to construct resistance matrices that represent the probabilities of gene flow between all pairs of populations. In areas of low resistance, movement of genes between populations is more likely, while high-resistance areas represent geographic and environmental barriers (Cushman et al., 2006; McRae, 2006). This methodological approach is essential to test more realistic hypotheses of gene flow (Isolation by Environment, IBE) (Orsini et al., 2013; Sexton, Hangartner & Hoffmann, 2014). However, to our knowledge, there have been few attempts to identify environmental barriers to gene flow of invasive species in their native range (Sherpa et al., 2019; Acevedo-Limón et al., 2020; Poveda-Martínez et al., 2023). This kind of evidence is essential for population management as input for invasion dynamic modeling to predict the expansion range in non-native regions (Brown et al., 2016; Aguirre-Liguori, Ramírez-Barahona & Gaut, 2021; Pilowsky et al., 2022).

The invasive cactus moth, Cactoblastis cactorum (Berg) (Pyralidae: Phycitinae), offers a unique opportunity to evaluate environmental barriers in the native range of an invasive species because inhabits a wide range of environmental conditions. Cactoblastis cactorum is a cactophagous species distributed in tropical and subtropical regions in South America, between 0 and 1,200 masl in Uruguay, south of Paraguay and Brazil, and in the central and northern part of Argentina (Mann, 1969; McFadyen, 1985; Varone et al., 2014), comprising the Chaco and Pampean biogeographical provinces (Morello et al., 2012; Oyarzabal et al., 2018; Arana et al., 2021; Morrone et al., 2022). Within this area, it uses several native host species of prickly pear cacti (O. megapotamica, O. elata, O. anacantha, O. bonaerensis, O. cardiosperma, O. surphurea, O. quimilo, O. rioplatensis, O. penicilligera) and the exotic O. ficus-indica (Marsico et al., 2010; Varone et al., 2014). The life cycle encompasses a gregarious larval stage within the cladodes, a pupal stage in the soil (approximately 5–10 cm in depth) and a free adult stage (Andraca-Gómez, 2011, personal observation). The whole cycle lasts between 4–5 months and depends on environmental conditions (Dodd, 1940; Pettey, 1948; Mann, 1969). In particular, temperature determines the percent of hatches (Legaspi & Legaspi, 2007; Marti & Carpenter, 2008).

This insect was initially used as a biological control agent against Opuntia in Australia, South Africa, and the Caribbean (Zimmermann, Bloem & Klein, 2007). After being introduced in the Caribbean in 1956 (Simmonds & Bennett, 1966), the cactus moth was dispersed to North America via commercial transportation and hurricanes (Simonsen, Brown & Sperling, 2008; Marsico et al., 2010; Andraca-Gómez et al., 2015, 2020), entering Florida in 1989, and since then, representing a major threat to the biodiversity and commercial production of Opuntia in Mexico (Soberón, Golubov & Sarukhán, 2001). Mexico is known to be one of the highest cactus biodiversity hotspots worldwide, as well as one of the main producers of Opuntia. Therefore, identifying environmental conditions that constrain the presence of C. cactorum in its native range can guide research on introduced ranges.

Previous studies in the native region (Argentina) using insect samples from seven host species of Opuntia revealed the presence of four genetic groups based on mitochondrial DNA (COI) (Marsico et al., 2010). Morphological differentiation of larvae was detected among the four genetic groups, which also were associated with different host usage, suggesting a possible host effect on ecotypic differences (Brooks et al., 2014). Although some degree of preference to oviposit on the exotic O. ficus-indica rather than on other native species was recorded, C. cactorum behave as a generalist with little host preference (Varone et al., 2014). Recent analyses using genome wide SNPs and niche modelling data indicated that past climatic changes during the Quaternary and shifts in host use conditioned the actual distribution of genetic variation of C. cactorum in Argentina (Poveda-Martínez et al., 2023). Ecological niche modelling using bioclimatic variables indicated that environmental suitability increases since the last glacial maximum (ca. 21 ky) from the west to the east, north and south of the present distribution (Poveda-Martínez et al., 2023). During the Spanish settlement in South America, five centuries ago, O. ficus-indica was introduced and likely colonized by C. cactorum since then (Ervin, 2012). The genetic structure of C. cactorum estimated across seven native hosts species suggest no evidence that the introduction of O. ficus-indica in the native range and the subsequent human-commercial dispersal have promoted contemporary admixture between distant populations (Poveda-Martínez et al., 2023). Within Argentina, O. ficus-indica occupies a larger area and a wider environmental range than any of the other native Opuntia species (Varone et al., 2014), representing a suitable system to examine possible contemporary environmental effects on genetic variation and structure without strong historical effects nested within native hosts distribution (e.g., Poveda-Martínez et al., 2023). To control these sources of variation and to explore the contemporary environmental factors that affect the genetic structure of the species, in the present study, species-specific nuclear microsatellites were used to characterize the geographic pattern of genetic variation in C. cactorum associated with the distribution of O. ficus-indica.

Genetic analyses were combined with ecological niche modelling to test the hypothesis that environmental conditions affected the genetic structure of the species. Given that the insect pupates in the upper soil layer (Zimmermann, Moran & Hoffmann, 2004) and is sensitive to temperature (Legaspi & Legaspi, 2007), we estimated its niche using soil and climatic variables to identify environmental barriers to species distribution. In addition, incorporating soil information in ecological niche models is known to reduce overestimation of expected suitability (Coudun et al., 2006; Beauregard & de Blois, 2014). The predictive model was used to build the Isolation by Environment (IBE) hypothesis represented by the resistance matrix to gene flow between pairs of sampling sites. A significant correlation between resistance and genetic differentiation matrices would indicate the existence of environmental barriers limiting dispersal (Hernández-Leal et al., 2022).

In the present study, we identified geographic and environmental (bioclimatic and soil) characteristics that may function as barriers for gene flow. Specifically, we (1) determined the existence of a significant genetic structure within the sampled region of Argentina where C. cactorum is associated with O. ficus-indica, (2) identified climatic and soil variables within the sampled region that better explain the distribution of C. cactorum following a niche modeling approach, and (3) combined these two pieces of evidence to test whether environmental conditions explain the geographic pattern of genetic differentiation (McRae, Shah & Mohapatra, 2009; Andraca-Gómez et al., 2015; Borja-Martínez et al., 2022).

Methods

Data collection

Between 2011 and 2012, 508 larvae were collected from 24 sites within the distribution range of C. cactorum in Argentina; mainly in the Chaco and Pampa biogeographic provinces and included three ecoregions (sampling approved by the Servicio Nacional de Sanidad y Calidad Agroalimentaria from Argentina) (Table 1, Fig. 1, Löwenberg-Neto, 2014). During two consecutive years, between February and March, one larva per cladode was collected, georeferenced and deposited in 1.5 ml vials with alcohol (96%) until DNA extraction. The samples were collected in the widely distributed exotic host, O. ficus-indica. Since this species was introduced five hundred years ago in South America, it is likely that it lacks a defensive mechanism against the cactus moth. Unlike native host species, this source of variation in the exotic host is minimized increasing the chance to examine environmental effects on the genetic structure of the cactus moth. Sample sizes varied between 10 and 30 individuals per site (Table 1, Fig. 1). DNA extraction was performed with the DNEasy® blood & tissue kit (cat.60504; QIAGEN, Germantown, MD, USA) and the resulting product was diluted to 20 ng/μl to warrant PCR amplification. We used microsatellites specifically developed for C. cactorum (Andraca-Gómez et al., 2020). The resulting PCR products were sent to the Core DNA Sequence Facility at the University of Illinois and analyzed in an Applied Biosystems sequencer (3,730 xl). The GeneMarker program (version 2.20 demo) was used to genotype individual larvae.

Table 1 List of 24 sampling sites of Cactoblastis cactorum in Argentina.

Sampling sites	Biogeographic province	Ecoregion	Political province	Coordinates	Number of individuals	
1. Huasapampa	Chacoan	Dry Chaco	Catamarca	27°54.839′S 65°33.805′O	30	
2. Icaño	Chacoan	Dry Chaco	Catamarca	28°55.996′S 65°17.955′O	27	
3. Hipódromo las Rosas	Chacoan	Dry Chaco	Catamarca	28°33.111′S 65°44.912′O	20	
4. Recreo	Chacoan	Dry Chaco	Catamarca	29°16.346′S 65°04.230′O	24	
5. San Martín	Chacoan	Dry Chaco	Catamarca	29°13.239′S 65°46.299′O	18	
6. El Talar	Chacoan	Dry Chaco	Catamarca	28°05.028′S 65°18.595′O	30	
7. San Isidro	Chacoan	Dry Chaco	Córdoba	32°08.933′S 65°06.292′O	19	
8. Cruz del Eje	Chacoan	Dry Chaco	Córdoba	30°42.304′S 64°48.602′O	30	
9. El Fortín	Pampean	Espinal	Córdoba	31°57.88′S 62°19.721′O	21	
10. Quilino	Chacoan	Dry Chaco	Córdoba	30°13.655′S 64°28.928′O	30	
11. Las Varillas	Pampean	Espinal	Córdoba	31°51.463′S 62°43.197′O	19	
12. Ayuí	Pampean	Espinal	Entre Ríos	31°11.727′S 58°02.797′O	15	
13. Federal	Pampean	Espinal	Entre Ríos	30°55.835′S 58°46.396′O	16	
14. Yuquerí	Pampean	Espinal	Entre Ríos	31°22.917′S 58°07.718′O	15	
15. El Carmen	Chacoan	Yungas	Jujuy	24°19.764′S 65°14.988′O	30	
16. Sastre	Pampean	Espinal	Santiago del Estero	31°44.344′S 61°50.193′O	10	
17. El Cuarenta y Nueve	Chacoan	Dry Chaco	Santiago del Estero	29°02.934′S 63°57.510′O	30	
18. Hock	Chacoan	Dry Chaco	Santiago del Estero	28°21.299′S 64°19.046′O	30	
19. La Banda	Chacoan	Dry Chaco	Santiago del Estero	27°44.937′S 64°12.232′O	11	
20. La Puerta	Chacoan	Dry Chaco	Santiago del Estero	27°37.915′S 64°37.281′O	26	
21. Pozo Escondido	Chacoan	Dry Chaco	Santiago del Estero	29°28.253′S 63°39.135′O	30	
22. Ruta Nueve	Chacoan	Dry Chaco	Santiago del Estero	27°45.027′S 64°23.532′O	12	
23. Vilmer	Chacoan	Dry Chaco	Santiago del Estero	27°45.982′S 64°09.632′O	15	
24. Tucumán	Chacoan	Dry Chaco	Tucuman	27°07.269′S 64°55.704′O	26	

Figure 1 Geographic location of the 24 sampling sites of Cactoblastis cactorum used for genetic analyses, genetic groups and barriers.

Samples are distributed in the Chacoan and Pampean biogeographic provinces (Löwenberg-Neto, 2014). The numbers correspond to those of Table 1. The six genetic groups defined by GENELAND are indicated in colored dots. Sampling sites: 9, 11, 12, 13, 14, 16 (green dots), 1, 6, 18, 20, 22, 23, 24 (yellow dots), 7, 8, 10 (blue dots), 17, 21 (purple dots), 2, 3, 4, 5 (red dots), Letters correspond to Salinas Grandes, SG, Salinas de Ambargasta, SA, Laguna Mar Chiquita, LA, Sierra de Ancasti, MN, Sierra de Ambargasta, MN, Sierra de Sumampa, MS, Sierra Grande, MG. Brown lines indicate the geographic location of the barriers proposed by the BARRIERS program (the barriers depicted are those with a percentage of existence greater than 70% after bootstrapping 100 random FST matrices). (B) Output of GENELAND analysis of the number of genetic clusters obtained from the 10,000 iterations with the larger likelihood (left). Analysis was performed with the uncorrelated allele frequency model option and 100,000 steps, thinning of 100, and burn-in of 100. The index of MCMC iteration indicate that Markov chains converged around six classes (genetic groups). Thus, the higher posterior probability was obtained for K = 6 in all 10 independent runs (right). (C) Observed heterozygosity (HO ± std) for each genetic group calculated as the average Ho for the 10 loci within each group (colored bars) and as the average Ho of sampling site within a given genetic group (white bars). (D) Matrix of paired genetic distances between genetic groups (all values are significant). The numbers and colors in Figures A, C and D are equivalent and represent the six genetic groups.

Genetic analyses

The presence of Hardy-Weinberg equilibrium and linkage disequilibrium at each location was tested with Genepop (web version, Rousset, 2008) while null alleles among loci were estimated with FreeNA. Loci with more than 20% of null alleles were eliminated from the analyses (Chapuis & Estoup, 2007), as well as those that were out of the Hardy-Weinberg equilibrium in more than 50% of the locations. FSTAT (version 2.9.3.2; Goudet, 2002) was used to calculate the number of alleles, the allele richness, the observed and expected heterozygosity, and differentiation between all pairs of sites and genetic groups (FST) (Weir & Cockerham, 1996; Chapuis & Estoup, 2007).

Genetic structure

First, a Bayesian grouping approximation was implemented in GENELAND (version 4.0) (Guillot, Santos & Estoup, 2008) in R Core Team (2023), to determine the existence of significant population genetic structure. GENELAND identifies groups of populations based on genetic similarity and geographic proximity. The analysis was performed in 10 independent runs of Monte Carlo Markov Chains (MCMC) with 100,000 chains, thinning of 100, burn-in of 100, and a minimum group value (K) of 1 and a maximum of 25. Assuming a significant genetic structure, uncorrelated allelic frequencies were chosen. We also incorporated the possible genetic ambiguity (excess homozygotes) in the grouping algorithm, assuming the existence of null alleles. The location of each individual in the analysis was included as a geographic coordinate in decimal degrees with a minimum distance of 0.001° (approximately equivalent to 100 m).

Second, to detect the presence of potential barriers to gene flow, we used the program BARRIERS (Version 2.2; Manni, Guerard & Heyer, 2004). This applies the Monmonier and Delaunay methods of triangulation of spatial coordinates of sampled sites and generates a map representing the relationship between the populations and the areas where the possible barriers can be found. We allowed a maximum of five barriers based on the number of genetic groups obtained by GENELAND. Genetic groups of populations were assigned a significance value after bootstrapping a set of 100 distance matrices using (Nei, 1972) genetic distance estimations. The 100 matrices required by the program were generated by resampling individuals within the populations using the program MSA (version 4.051). To examine the extent of genetic isolation of potential genetic groups a multivariate analysis of molecular variance (AMOVA) was performed to decompose the total amount of genetic variation among and within genetic groups (Arlequin 3.5; Excoffier & Lischer, 2010).

Ecological niche modeling and environmental barriers

To identify environmental barriers related to genetic grouping of sampled sites, niche modeling and isolation by resistance analyses were combined (Manthey & Moyle, 2015; McRae & Beier, 2007; McRae et al., 2008). The MaxEnt algorithm executed in the ntbox package in R (Osorio-Olvera et al., 2020) was used to build a niche model hypothesis for the sampled area of C. cactorum. To carry out the modeling, we used 40 sites in Argentina where individuals of C. cactorum were observed during sampling. To build the model, climatic and soil variables were gathered from WorldClim (https://www.worldclim.org/data/bioclim.html), Soil (Biosoil) (https://zenodo.org/record/4558732) (Lembrechts et al., 2022) and SoilGrids (https://www.isric.org/explore/soilgrids) databases. We curated our occurrence data using standard steps in ecological niche modeling literature and using the approach of Cobos et al. (2018). We eliminated spatial duplicates by using a threshold distance of 0.04 grades (~2.5 km at the equator). To avoid collinearity-related problems, we estimated the correlation among each pair of predictors and kept only those with correlation values <0.7. We ran iteratively MaxEnt models using its auto features and explored variable contribution via the Jackknife test on AUC values (area under the receiver operating characteristic (ROC) curve). After each run, we removed the least contributing variable from the list of non-correlated environmental variables. After the selection model procedure, using AUC, we ended up with the six best environmental variables that had the highest contribution in most of the models. The final model prediction (suitability map) expressed as a raster file was used in CIRCUITSCAPE (version 4.0, McRae, Shah & Mohapatra, 2009) to construct the resistance matrix (Andraca-Gómez et al., 2020). Geographic points with low suitability delineate areas of high resistance for establishment, suggesting the presence of a geographic or environmental barrier. Multiple matrix regression with randomization (MMRR) was performed using the genetic distance matrix based on FST/(1 − FST) values between pairs of sites as the response variable against the geographic distance matrix (log) and the resistance (environmental) matrix obtained in CIRCUITSCAPE following the niche model prediction (Wang, 2013). The distance matrix was adjusted to control for the great-circle distance (i.e., shortest distance between two points on the surface of a sphere) using the package sf in R (Pebesma, 2018). The model parameters of the multiple regression were obtained after 999 random permutations of rows and columns of the dependent genetic distance matrix to generate a null distribution against which observed values were contrasted (Legendre, Lapointe & Casgrain, 1994).

Results

Genetic variation and structure. After an initial study, four out of 14 nuclear microsatellite loci were eliminated because they had a null allele frequency greater than 20%. A total of 10 microsatellites comprising 152 alleles were used in the final analyzes (https://doi.org/10.6084/m9.figshare.24749082). Among the 24 locations sampled, the allele richness varied between 3.36 and 5.78 and the observed heterozygosity (Ho) between 0.36 and 0.63 (Table 2). All sites, except site 14 (Yuquerí), had fewer heterozygotes than expected under the Hardy-Weinberg equilibrium (FIS > 0, Table 2). Significant paired genetic differentiation among sites ranged from FST = 0.0228 between locations 22 and 24 to FST = 0.3011 between locations 4 and 12. The mean level of genetic differentiation for the whole set of sampling sites was FST = 0.178. Within the sample region, the analysis of genetic structure using GENELAND indicated that the most probable number of genetic groups (k) was six (Fig. 1B). Genetic groups (hereafter populations) were defined by a probability of assignment between 0.30 and 0.36 (Fig. 1A). The 15th collection site corresponds to an isolated group in the northern Yungas ecoregion, within a mountain forest near the Dry Chaco. On the east side of the distribution, within the Pampean province, there is a group of six sampling sites (green dots in Fig. 1A) corresponding to the Espinal ecoregion with humid flats between the Paraná and Uruguay rivers. On the west area of the distribution within the Dry Chaco ecoregion, there are four genetic groups: a northwestern group (yellow dots in Fig. 1A), a southwestern group (blue dots in Fig. 1A), and two groups in the middle, one on the east border (purple dots in Fig. 1A) and another on the west border (red dots in Fig. 1A). The results of AMOVA indicated that the variation within sites accounted for most of the genetic variation (81.8%) followed by the variation among sites within genetic groups (9.9%) and the variation among genetic groups (8.26%). Genetic differentiation among genetic groups was FCT = 0.078 (Fig. 1D). Heterozygosity for each genetic group estimated using the pooled sample of sites was similar to the average Ho when using each site as a replicate (Fig. 1C). The presence of potential barriers to gene flow with a probability of more than 50% existence strongly matched the clustering proposed by GENELAND (Fig. 1A). The barriers with higher probability delimited the four genetic groups within the west region of the distribution range, while less intense barriers separated the north and east regions (Fig. 1A). Clusters 1, 2, 3, and 5, correspond to the Dry Chaco ecoregion, while cluster 6 corresponds to the Yungas ecoregion close to the Dry Chaco. Cluster 4 is located within the Pampean province, in a humid flat, within the Espinal ecoregion. Clusters 1, 2, 3, and 5 within the Dry Chaco are separated by mountain ranges, salt flats, and wetlands in arid or semi-arid conditions. Group 1 in the north (yellow dots in Fig. 1A) corresponds to forests and shrublands, to the north of Salinas Grandes and south of the wetlands of the Salado river. Group 2 is located in salt flats within the Monte ecoregion surrounded by the Sierra de Ancasti to the north and Salinas Grandes to the west (red dots in Fig. 1A). Group 3 corresponds to dry forests and shrublands in a zone of low mountains, south of Salinas Grandes and west of Sierra Grande (blue dots in Fig. 1A). Group 5 is located within an area surrounded by Salinas de Ambargasta (East), Sierra de Ambargasta and Sierra de Sumampa (South), Salina del Saladillo (North) and delta of the Dulce River and Mar Chiquita (National Park Ansenuza Lagoon (Northeast) (black dots in Fig. 1A).

Table 2 Statistics of genetic diversity of Cactoblastis cactorum in 24 sampling sites from Argentina for 10 nuclear microsatellite loci.

Sampling sites	NA	AR	HS	HO	FIS	
1. Huasapampa	6 (2.494)	4.736 (1.799)	0.684 (0.215)	0.491 (0.222)	0.282	
2. Icaño	5.7 (1.16)	4.503 (0.992)	0.626 (0.194)	0.444 (0.188)	0.291	
3. Hipódromo las Rosas	4.9 (2.132)	3.940 (1.458)	0.560 (0.238)	0.41 (0.251)	0.272	
4. El Recreo	4.8 (2.251)	3.834 (1.646)	0.538 (0.240)	0.365 (0.180)	0.323	
5. San Martín	4.1 (1.524)	3.773 (1.170)	0.578 (0.188)	0.448 (0.218)	0.231	
6. El Talar	5.9 (2.685)	4.789 (1.841)	0.655 (0.250)	0.427 (0.191)	0.348	
7. San Isidro	5.4 (2.413)	4.556 (1.879)	0.628 (0.270)	0.377 (0.274)	0.399	
8. Cruz del eje	7.2 (2.57)	5.508 (1.870)	0.708 (0.219)	0.487 (0.161)	0.313	
9. El Fortín	5.9 (1.595)	4.742 (1.247)	0.670 (0.184)	0.551 (0.177)	0.177	
10. Quilino	6.3 (2.058)	5.133 (1.548)	0.695 (0.209)	0.458 (0.214)	0.207	
11. Las Varillas	6.8 (1.814)	4.934 (1.100)	0.681 (0.098)	0.515 (0.140)	0.247	
12. Ayuí	3.8 (1.687)	3.365 (1.349)	0.533 (0.235)	0.441 (0.244)	0.171	
13. Federal	5.4 (1.578)	4.649 (1.301)	0.664 (0.187)	0.526 (0.233)	0.207	
14. Yuquerí	4.2 (1.619)	3.732 (1.237)	0.586 (0.243)	0.560 (0.278)	0.052*	
15. El Carmen	6.8 (1.814)	5.008 (1.371)	0.653 (0.188)	0.508 (0.189)	0.225	
16. Sastre	5.5 (1.581)	5.333 (1.597)	0.645 (0.232)	0.474 (0.210)	0.275	
17. El Cuarenta y nueve	6.7 (2.003)	5.468 (1.611)	0.722 (0.174)	0.417 (0.190)	0.423	
18. Hock	6.2 (1.687)	4.967 (1.182)	0.730 (0.133)	0.440 (0.194)	0.402	
19. La Banda	4.4 (1.174)	4.219 (1.157)	0.608 (0.135)	0.427 (0.268)	0.307	
20. La Puerta	6.1 (1.912)	4.914 (1.310)	0.703 (0.134)	0.582 (0.148)	0.175	
21. Pozo Escondido	7.1 (2.685)	5.502 (1.611)	0.719 (0.176)	0.459 (0.251)	0.365	
22. Ruta 9	6 (2.211)	5.577 (1.986)	0.725 (0.213)	0.633 (0.261)	0.131	
23. Vilmer	5.4 (1.713)	4.720 (1.399)	0.692 (0.150)	0.477 (0.229)	0.319	
24. Tucumán	6.8 (2.15)	5.458 (1.483)	0.719 (0.132)	0.541 (0.223)	0.251	
Note:

Number of alleles (NA), allelic richness (AR) (estimated from nine diploid individuals), expected heterozygosity (HE), observed heterozygosity (HO), inbreeding coefficient ((FIS (*non-significant values)).

Niche modeling. The niche model of C. cactorum had an AUC value of 0.875 and an omission rate of zero under a five percentile threshold corresponding to a suitability value of 0.074. The main environmental variables that better explained the distribution of the moth were related to precipitation and temperature on the soil surface and within the upper soil layer (10 cm depth), as well as the soil carbon content. These correspond to: average temperature of the driest quarter (relative contribution to the model, 30%), maximum soil temperature of the warmest month (relative contribution to the model, 16.1%), annual temperature range (relative contribution to the model, 14.6%), precipitation seasonality (relative contribution to the model, 14.3%), mean soil temperature of the wettest quarter (relative contribution to the model, 13.7%), and soil organic carbon density (relative contribution to the model, 9.9%). A higher environmental suitability was detected in the west region where more genetic groups were found. From the west to the north and east areas of the distribution, the environmental suitability declines consistently (Fig. 2).

Figure 2 Suitability map for Cactoblastis cactorum as predicted by the consensus niche model (AUC = 0.875).

The best model had an omission rate of zero under a five percentile threshold corresponding to a suitability value of 0.074 (https://doi.org/10.6084/m9.figshare.24749082). Colors indicate the model predicted suitability within the sampled region. Regions with high suitability indicate a higher probability of detecting C. cactorum in Opuntia ficus-indica.

Environmental-genetic association. The MMRR analysis showed that the environmental distance matrix (based on the prediction of the niche model) was significantly related to the genetic distance matrix (βE = 0.506, P = 0.032) supporting the hypothesis of Isolation by Environment (IBE). On the contrary, the same analysis rejected the hypothesis of Isolation by Distance (IBD) (βD = 0.053, P = 0.793) (that is, there is no significant association between genetic and geographic distance matrices. The data better support the hypothesis of environmental filters influencing the genetic structure and dispersal of C. cactorum than geographic distance.

Discussion

Among plant-natural enemy interactions, environmental conditions and host species affect the distribution of the genetic variation of consumers (Mopper & Strauss, 2013; Whitham et al., 2003; Wang & Bradburd, 2014; Wang et al., 2017a). Our analyses demonstrate the existence of a significant genetic structure of C. cactorum in Argentina associated with soil and climatic variables besides the presence of the exotic host O. ficus-indica (introduced in this region about 500 years ago). While the western part of the distribution comprises more genetic diversity (four genetic groups) and has higher environmental suitability, the genetic groups in the east and north correspond to areas with lower environmental suitability. The environmental suitability of the western region corresponds to an area with high environmental heterogeneity (Oyarzabal et al., 2018) but climatically more stable during the Quaternary (Poveda-Martínez et al., 2023) representing a Pleistocene refuge for biodiversity during the last glaciation (Baranzelli et al., 2017; Robbiati et al., 2021; Camps et al., 2018). Furthermore, the suitability for C. cactorum in the sampled region seems to be highly influenced by temperature and precipitation above and below ground, in combination with other soil characteristics. Genetic analyses, allowed us to identify barriers corresponding to mountain ranges, salt flats, wetlands, and the largest lagoon in central Argentina (Mar Chiquita). These barriers delimited areas with significant variation in temperature and precipitation that influenced the genetic clustering of prickly pear moth populations and may represent major environmental filters for its distribution, dispersal, and genetic variation.

The levels of genetic diversity estimated by heterozygosity showed deficiency (FIS > 0) in most of the samples of C. cactorum, excepting sampling site 14 (Yuquerí). Deficiency of heterozygotes and a high proportion of null alleles (>20%) are a common phenomenon among Lepidoptera (Malausa et al., 2007; Sinama et al., 2011; Guillemaud et al., 2015). This condition is associated with high rates of mutation in genetic regions flanking microsatellites, as well as the presence of transposable elements (Sinama et al., 2011). Other factors like gene flow, genetic drift, and the genetic structure of populations (Wahlund effect) can also account for lower-than-expected levels of heterozygotes (Haldane, 1948; Kimura, 1968). When the average heterozygosity for each genetic group was compared with the observed heterozygosity for the entire genetic group, no differences were observed. This suggests that possible Wahlund effects were not likely related to the genetic structure of populations (Waples, 2015). The heterozygosity was rather uniform among the sampling sites, suggesting that there were no strong effects of genetic drift. Furthermore, the east genetic group had the lowest FIS values and is less differentiated from the other groups. Despite significant paired genetic differentiation between sampling sites, the low amount of variance explained by genetic groups suggests that gene flow has been moderate. Levels of paired genetic differentiation among sampling sites (range FST = 0.022–0.301) fall within the range detected using nuclear SNPs across a pooled sample of seven hosts within the same region (FST = 0.023–0.448) (Poveda-Martínez et al., 2023). Ongoing genomic analyses will provide more information on selection pressures, demographic history and potential barriers to gene flow to explain positive FIS values and to unravel the intricate mechanism shaping genetic variation in the cactus moth.

Our results indicate the presence of a significant genetic structure of the cactus moth on the exotic O. ficus-indica, a species introduced about five centuries ago during the Spanish arrival to South America (Ervin, 2012). The recent history of the host shift to O. ficus-indica suggests that the environmental heterogeneity within the sampled region plays a more important role than the host on the genetic structure of the cactus moth. This is further supported because since its introduction to South America, O. ficus-indica likely had little chance to evolve specific defensive mechanisms against the cactus moth. The west sampled region (within the Dry Chaco) contained the highest genetic diversity and suitability represented by four genetic groups (1, 2, 3, 5), which are delimited by mountain ranges, salt flats, and wetlands in arid or semi-arid conditions. This finding mirror previous research indicating that Dry Chaco corresponded to a biodiversity refuge during the Quaternary climate changes (Poveda-Martínez et al., 2023), and suggest an association between genetic diversity and environmental suitability (Ochoa-Zavala et al., 2022). Colonization of C. cactorum to O. ficus indica followed an historical phylogeographic pattern seen in other species, promoted by more recent environmental conditions. This is supported by two previous findings: (1) the generalist feeding habit of the cactus moth (Varone et al., 2014) that likely allowed the colonization of O. ficus-indica since its introduction, (2) the absence of a long coevolutionary history of O. ficus-indica and the cactus moth, and (3) the absence of human-mediated dispersal of O. ficus-indica related to agroindustry that promote admixture among distant populations (Poveda-Martínez et al., 2023). Since its introduction in the Dutch Antilles in 1956 (Simmonds & Bennett, 1966), a similar pattern was found in the invaded region of North America (Florida) and the Caribbean (Andraca-Gómez et al., 2020), where the moth followed the phylogeographic pattern recorded for other native species of turtles, birds, crabs, and beetles (Avise, 2000). Thus, the presence of C. cactorum on O. ficus-indica in Argentina represents a useful natural setup to disentangle the effect of the host and the environment in a species that interact with various hosts inhabiting different environmental conditions (Wang et al., 2017b).

Ecological niche models in herbivorous insect species have shown that the host plays an important role in their distribution range (Giannini et al., 2013; Simões & Peterson, 2018). For example, an important improvement in the model performance was detected for the tortoise beetle Eurypedus nigrosignatus when including host information in their niche models. Besides the presence of the host species, our results indicate that temperature (above and below ground), precipitation (seasonality), and soil organic carbon content can be the most relevant variables to predict the distribution of the cactus moth in the sampled region. Our results add to previous results of niche modeling for C. cactorum in North (Soberón, Golubov & Sarukhán, 2001) and South America (Poveda-Martínez et al., 2023) using only bioclimatic variables as soil characteristics significantly contributed to the model prediction. Since the moth pupates approximately in the top 10 cm of soil, temperature below the growth level, moisture and organic carbon content probably play a major role in pupal survival. Other species of lepidopteran have a high mortality rate during the pupal stage when soil humidity increases (Wang et al., 2017b; Shi et al., 2021; Tian et al., 2021), but a low content can also affect pupal survival and emergence (Wang et al., 2017b). Experimental studies and demographic analyses in different populations of C. cactorum in South Africa and under experimental conditions in Florida, found a lower development of larvae at <18 °C and >34 °C (Zimmermann & Moran, 1991; Legaspi & Legaspi, 2007). In the present study, the greater environmental suitability in the drier western region suggests that pupae are probably more vulnerable to high soil moisture during the summer as precipitation is drastically reduced from the eastern plains of the Pampean region to the semi-arid shrublands and dry forests of Dry Chaco (Oyarzabal et al., 2018). The lower number of populations and the environmental suitability of the eastern group support the expectation that this region is under less benign conditions for moth development on O. ficus-indica. Ecological niche theory proposes that more populations will be found at the center of the ecological niche (Martínez-Meyer et al., 2013; Osorio-Olvera et al., 2020), corresponding to the area with optimal conditions for survival, growth, and reproduction (Lira-Noriega & Manthey, 2014; Osorio-Olvera et al., 2016). Our results support this expectation, as the region with higher environmental suitability following the niche model also corresponds to the region where C. cactorum was more abundant and where more genetic groups were detected. As environmental suitability is not homogeneously distributed within the sampled region, patterns of dispersal and genetic differentiation would be affected by environmental filters (e.g., Acevedo-Limón et al., 2020; Valdez, Quiroga-Carmona & D’Elía, 2020; Hernández-Leal et al., 2022).

In particular, the isolation by environment hypothesis (IBE) following the principles of electric resistance has helped to identify potential environmental barriers to species distribution and gene flow (McRae, 2006; Wang & Bradburd, 2014). This approximation has increased the predictive power to account for the spatial distribution of genetic variation (McRae, Shah & Mohapatra, 2009; McRae & Beier, 2007; Wang & Bradburd, 2014; Andraca-Gómez et al., 2015). Whereas the IBE hypothesis can be constructed using natural history information, niche models can provide a quantitative more precise estimation of environmental suitability (see Andraca-Gómez et al., 2015 and Poveda-Martínez et al., 2023). The significant effect of the environment on the distribution of genetic variation allowed us to successfully identify important geographic and environmental barriers for gene flow and/or genetic differentiation in C. cactorum. Our results extend previous findings that the central Dry Chaco region comprises the ancestral genetic lineage (Poveda-Martínez et al., 2023), indicating that this area also present high diversity of genetic groups and the presence of significant environmental barriers. One of the strongest barrier separated the westerns groups within the Dry Chaco from sites located in the Pampean province (e.g., Poveda-Martínez et al., 2023). Barriers represented by mountain ranges, salt flats, wetlands, and soil conditions translate to different combinations of humidity and temperature of the upper soil layer where the moth pupates. Therefore, this stage of the life cycle seems to be critical for the environmental tolerance of the moth. Although the presence of a suitable host is a prerequisite for survival, it is not a sufficient condition for the presence of C. cactorum. In fact, during sampling, the moth was not detected at several sites where O. ficus-indica was present (Andraca-Gómez, 2011, personal observations). Given the climatic and soil differences among the genetic groups, phenological asynchrony is expected, reducing the opportunities for effective gene flow (Zimmer & Emlen, 2013) and probably a higher heterogeneity in the life history traits of the cactus moth. This may explain the presence of at least four genetic groups within the western region. Overall, our results provide a new piece of evidence to understand the relevance of contemporary environmental conditions on the genetic structuring of this invasive species within its native range.

Supplemental Information

Supplemental Information 1 Genetic differentiation values (FST) between pairs of sampling sites.

Table S1.Genetic differentiation values (FST) between pairs of sampling sites.

Click here for additional data file.

Supplemental Information 2 Raw data of nuclear microsatellite information of the sampled individuals of Cactoblastis cactorum in Argentina.

Click here for additional data file.

The authors thank Paula Zamudio and collaborators (Fundación Miguel Lillo) during field trips and insect maintenance in the lab, and to Marco Tulio Solano de la Cruz and Rubén Pérez-Ishiwara for technical assistance. Ella Vázquez, Travis Marsico and two anonymous reviewers provided constructive comments to the final version of the manuscript.

Additional Information and Declarations

Competing Interests

Author Contributions

Field Study Permissions

Data Availability

The authors declare that they have no competing interests.

Guadalupe Andraca-Gómez conceived and designed the experiments, performed the experiments, analyzed the data, prepared figures and/or tables, authored or reviewed drafts of the article, and approved the final draft.

Mariano Ordano conceived and designed the experiments, performed the experiments, authored or reviewed drafts of the article, and approved the final draft.

Andrés Lira-Noriega analyzed the data, authored or reviewed drafts of the article, and approved the final draft.

Luis Osorio-Olvera analyzed the data, authored or reviewed drafts of the article, and approved the final draft.

César A. Domínguez conceived and designed the experiments, authored or reviewed drafts of the article, and approved the final draft.

Juan Fornoni conceived and designed the experiments, authored or reviewed drafts of the article, and approved the final draft.

The following information was supplied relating to field study approvals (i.e., approving body and any reference numbers):

Servicio Nacional de Sanidad y Calidad Agroalimenaria (Argentina).

The following information was supplied regarding data availability:

The data is available at figshare: Andraca-Gómez, Guadalupe (2023). Script and database. figshare. Dataset. https://doi.org/10.6084/m9.figshare.24749082.v1.

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
