# Peer review of "Climatic and soil characteristics account for the genetic structure of the invasive cactus moth Cactoblastis cactorum, in its native range in Argentina"

_PeerJ, doi:10.7717/peerj.16861_

## Round 0.1 · original submission · Major Revisions

Dear Authors, as you will see in the Reviewers' comments - the study got favourable comments, but reviewers identified some issues that need fixing. In addition to their comments I would also suggest revising the paper in terms of typos (e.g., the way citations are reported - see e.g. L188, or obvious mistakes - like burning instead of burn-in, L196).

Other comments:
- Can you please add details on MCMC chain convergence (how it was tested and also to confirm chains did converge)
- L240-241 - so you had 10 loci, what is the 152 number added here? (it says "loci" but then the sentence is contradictory).
- L243 - make it H_lower-index-0, not Ho
- Figures 1 and 2 - the two figures are redundant, figure 2 repeats all geographic information from Fig 1, so it could easily be omitted (just adding population IDs to Fig 2) - it will also make it more information dense and will reduce the need of jumping between 2 figures.

Reviewer 1 ·

Basic reporting

The manuscript, PeerJ_90876, makes a valuable contribution by aiming to identify the population genetic structure of the South American cactus moth, Cactoblastis cactorum. The literature review conducted by the authors is comprehensive, well-referenced, and includes recent genetic studies on this species in the region. While the figures are relevant, I recommend improving the labeling and descriptions for better clarity. Clearly labeled figures enhance reader understanding and interpretation of the data. Providing more detailed captions can also contribute to a smoother flow of information. I appreciate the inclusion of raw data, as it is crucial for reproducibility. The English employed in the manuscript is appropriate and unambiguous, contributing to the overall clarity of the text. This enhances the accessibility of the research to a wider audience. Feel free to adapt these suggestions based on the specific aspects you want to emphasize or the tone you'd like to convey in your review.

Experimental design

In my assessment, the experimental design is sound, and the authors have clearly defined their research questions. The investigation was conducted with high technical and ethical standards, employing 10 nuclear microsatellites to characterize the genetic structure across 24 sampling locations in Argentina. The inclusion of a substantial number of individuals and the application of various genetic population analyses are adequate. However, to enhance reproducibility, I recommend providing more detailed information in certain sections of the methods (mentioned below). Including these additional details would strengthen the transparency and reproducibility of the methodology.

Validity of the findings

I strongly recommend that the authors consider making the genetic data generated in this study available in established repositories.

Additional comments

L151 Please check the correct scientific names of host plants. O. anaconda??; change Opuntia surfurea to O. sulphurea

L145 - 158 I recommend incorporating this section into the introduction, specifically during the introduction of the species.

L163 Could the authors please provide the number or details of the permissions obtained for the sampling collections, if available?

L164 Please delete an extra parenthesis

L206 I would appreciate clarification on the specific objectives or rationale for conducting the AMOVA analysis in this study.

L228 I appreciate the inclusion of the MaxEnt-derived habitat suitability maps and subsequent analysis using Circuitscape in the manuscript. To enhance reproducibility and ensure a clear understanding of the methodology, could you please provide more detailed information on the specific steps taken to transform the MaxEnt output into a resistance matrix for use in Circuitscape? This may include details on thresholding, raster conversion, assignment of resistance values, and any custom scripts or parameters employed in the process. Clear documentation of these steps would greatly benefit readers aiming to reproduce or build upon your analysis.

L232 I would appreciate additional details regarding the MMRR analysis in the manuscript. Specifically, could you provide clarity on how the best fit was determined among the genetic distance matrix and explanatory terms? Additional information on model specifications, the randomization procedure, and any sensitivity analyses conducted would be beneficial for a comprehensive understanding of the analysis.

L250 - 279 I recommend using a consistent term, either 'genetic groups,' 'groups,' or 'populations' throughout the manuscript to enhance clarity and maintain uniformity. This will help avoid potential confusion for readers and ensure a smoother understanding of the text.

L303 I recommend modifying the first sentence in the Discussion section to avoid the impression that the genetic structure in C. cactorum is attributed to O. ficus-indica. It would be more accurate to emphasize that the observed genetic structure is likely influenced by various environmental conditions rather than solely by the presence of O. ficus-indica.

L336 While in the introduction and your results have discussed factors such as shifts in native host distributions and environmental influences as contributors to genetic differentiation in the cactus moth, it would be beneficial to elaborate on the need for new genomic analyses that you suggest in this point. Considering the complex interplay of factors in genetic differentiation, additional genomic analyses could provide a more comprehensive understanding. Exploring factors such as demographic history, selection pressures, and potential gene flow barriers might offer insights into the intricate mechanisms shaping genetic variation in the cactus moth. Discussing these aspects could strengthen the manuscript's contribution to the broader understanding of genetic differentiation in this species.

L341 -348 I recommend revising the discussion at this point. Given the relatively recent introduction of O. ficus-indica (500 years ago), it is unlikely to have induced significant evolutionary changes in lepidopterans within this timeframe. Instead, I suggest a more detailed exploration of the great environmental heterogeneity characterizing the region. Emphasizing the broader environmental context, including factors such as climate and habitat diversity, would provide a more nuanced interpretation of the genetic differentiation observed in C. cactorum. This would help avoid conclusions solely based on the distribution of the exotic host plant and contribute to a more comprehensive understanding of the ecological factors influencing genetic patterns in this species.

L350 Which geographic barriers did you identify? please clarify.

L365 - 368 and L396 - 402 These paragraphs in question appear to have an introductory tone. I recommend re-editing it to transition into a more focused discussion of your findings.

L706 I recommend considering the inclusion of Figure S1 in the main text rather than relegating it to the supplementary files. Placing the figure in the main text can enhance its visibility and accessibility for readers, providing a more integrated presentation of the supporting data within the flow of the manuscript.

Table 1. I recommend providing additional geographic details in the Table1, particularly specifying the provinces where each sample was collected. Please consider incorporating this detail in the maps Figure 1.

·

Basic reporting

The authors present a well-written article on the environmental factors that may structure populations of the South American Cactus Moth (Cactoblastis cactorum). There are only a few typographical and grammatical errors in the writing, and I am providing a marked up PDF with specific changes noted. The referenced literature is relevant to put the article into context, but the literature cited section currently suffers from inconsistencies in formatting. Again, see attached PDF for specific corrections, but note that I did not review each citation very carefully, and I expect the authors will do so to align formatting throughout the Literature Cited section. The tables and figures are appropriate. Supplementary Figure S1 needs an edit. The red rectangular box showing the location of the color map needs to be shifted to the right (east) to accurately depict the location of the area shown. I also suggest some reorganization of the Discussion section to bring the main points closer to the top of that section...specific recommendations in the attached.

Experimental design

The study is well designed and the design is justified. It was exciting to see the niche modeling variables used in the study and the linkage to the relevance to the moth's natural history. It seems worth noting, maybe somewhere in the methods, that researching the environmental variables for an herbivore feeding on an exotic host has the advantage of removing differential defenses from the host plants. In other words, an exotic suitable host likely has fewer defenses against feeding from C. cactorum, which allows the research team to tease out the environmental factors while controlling the biotic conditions.

Validity of the findings

The findings are valid and appear to be accurately and correctly reported. The conclusions are well stated and related to the objectives. The last sentence of the Results section reads as a little strong to me as a statement of cause and effect. Maybe the authors will consider something more like, "The data better support the hypothesis of environmental filters influencing the genetic structure and dispersal of C. cactorum than geographic distance."

---

## Round 0.2 · Minor Revisions

Thank you for submitting the revised version of your paper, it has now much improved and is basically ready for publication. I'm indicating minor revision to ask for corrections of few minor things:
L90: I think "this species" would sound much better than "this insect"
L171: lacks, not lack
L243: log instead of Log (lower-case)
L741: five-percentile (add dash)
L734: "y" used instead of "and"?

Reviewer 1 ·

Basic reporting

I have completed a thorough review of the revised manuscript, PeerJ_90876, and I am pleased to report that the authors have diligently addressed all the concerns raised during the initial review process. They have successfully incorporated the suggestions, comments, and recommendations provided by the reviewers, resulting in a significantly improved version of the manuscript.

After careful consideration, I believe that the authors have effectively addressed all the major issues, and the manuscript is now ready for acceptance. The revisions demonstrate a commitment to enhancing the clarity, rigor, and overall quality of the research presented.

Experimental design

.

Validity of the findings

.

Additional comments

.

·

Basic reporting

The authors have submitted a revision of their manuscript, addressing comments and concerns from two reviewers and the editor. Aside from minor subject-verb agreement in the new edits, the authors have done a great job addressing the concerns and getting the manuscript ready for publication. I have not further concerns, and as long as the minor typographical edits are fixed before publication, I think the manuscript is ready for publication.

Experimental design

The experimental design was good before, and now it is even further detailed in the revision. Concerns have been effectively addressed.

Validity of the findings

The findings are important and in a relevant context. With the minor edits to the discussion and improvements to the figures, the validity of the findings are even clearer than in the original submission.

Additional comments

I have attached the marked-up Word document to show a few very minor additional edits to be included before publication.

---

## Round 0.3 · accepted · Accept

Thank you for addressing all comments of myself and the referees. I assessed they way they were implemented and I'm happy to accept the paper for publication in PeerJ.